# A Single-Cycle Adenovirus Type 7 Vaccine for Prevention of Acute Respiratory Disease

**DOI:** 10.3390/v11050413

**Published:** 2019-05-03

**Authors:** Brianna L. Bullard, Brigette N. Corder, Eric A. Weaver

**Affiliations:** School of Biological Sciences, Nebraska Center for Virology, University of Nebraska, Lincoln, NE 68503, USA; bbullard@huskers.unl.edu (B.L.B.); brigette.corder@huskers.unl.edu (B.N.C.)

**Keywords:** adenovirus, type 7, vaccine, single cycle, military, pediatric deaths, acute respiratory disease, ARD

## Abstract

Adenovirus type 7 (Ad7) infection is associated with acute respiratory disease (ARD), especially in military recruits living in close quarters. Recently, several outbreaks of Ad7 infections have occurred in civilian populations, with some cases leading to death. However, the current Ad7 vaccine is licensed for use only in military recruits because it utilizes an orally delivered wild type virus which is shed in the stool for 28 days after immunization. This poses a safety risk due to the possibility of virus spread to vulnerable populations. To address the need for a safer Ad7 vaccine for use in civilian populations, we developed a single-cycle Ad7 virus (scAd7). This scAd7 virus is deleted for the Ad7 fiber protein, so that viruses produced outside of complementing cells lines lack this essential structural protein and have severely reduced infectivity. In vitro studies in noncomplementing A549 cells showed that the scAd7 virus has genomic DNA replication kinetics and Ad7 hexon expression similar to a replication-competent virus; however, virus progeny produced after infection has impaired infectivity. Therefore, this scAd7 virus combines the safety advantages of a replication-defective virus with the increased Ad7 gene expression of a replication-competent virus. Due to these advantages, we believe that scAd7 viruses should be further studied as an alternative, safer Adenovirus 7 vaccine.

## 1. Introduction

Adenovirus type 7 (Ad7) infection is associated with acute respiratory disease (ARD), fever, bronchitis, and pneumonia [1]. Young children, adults in crowded environments, and immunocompromised patients are more susceptible to increased disease severity [2]. Ad7, along with Adenovirus type 4 (Ad4), causes significant morbidity in military recruits. Since 1971, millions of military recruits have been vaccinated against Ad7 and Ad4 using an orally delivered, lyophilized wild-type virus, which has been found to be effective at reducing the rates of respiratory febrile illness [3,4,5]. However, this vaccine is only licensed for military populations because the wild type virus can be shed in stool for up to 28 days post-vaccination [5,6]. Vaccinees are advised to exercise caution during this time and limit contact with vulnerable populations, such as pregnant women, immunocompromised people, or young children [6].

The Ad7 vaccine is only available to military recruits; however, the general population is also at risk for severe infection. Recently, in 2018, 11 pediatric deaths were reported in a New Jersey hospital as a result of Ad7 infection [7]. That same year, 35 confirmed cases of Adenovirus infection, leading to 1 death, were reported at the University of Maryland, with Ad7 as the causative type in at least 10 of the reported cases [8]. Epidemic Ad7 infections cause significant morbidity worldwide in both military and civilian populations [9,10,11,12,13].

To address the need for a safe Ad7 vaccine for use in the general public, we developed a single-cycle Ad7 virus that combines the safety advantages of a replication-defective virus with the increased Ad7 gene expression of a replication-competent virus [14]. As a single-cycle virus, the Ad7 genome is replicated ~10,000 fold after viral entry, leading to greater expression of Ad7 viral proteins. However, the virus produced outside of the complementing cell line lacks the fiber protein. Fiber protein binding to cellular receptors is the primary mode of transduction for Adenovirus; therefore, this fiberless Ad particle has severely reduced infectivity [15,16]. This greatly increases the safety profile of this Ad7 vaccine while maintaining strong Ad7 protein expression, making it an ideal candidate to explore for prevention of respiratory disease caused by Ad7 infection.

## 2. Materials and Methods

### 2.1. Ethics Statement

Human desmoglein (hDSG) transgenic mice were generously gifted from Dr. André Lieber [17]. Breeding colonies were established under the University of Nebraska—Lincoln (UNL) IACUC protocol number 1243. Mice were housed in the Life Sciences Annex building on the UNL campus under the Association for Assessment and Accreditation of Laboratory Animal Care (AALAC) guidelines with animal use protocols approved by the UNL Animal Use and Care Committee under protocol number 1241 approved 11/19/2015. All animal experiments were carried out in accordance with the provisions of the Animal Welfare Act, PHS Animal Welfare Policy, the principles of the NIH Guide for the Care and Use of Laboratory Animals, and the policies and procedures of UNL. All immunizations and bleeds were performed under ketamine- and xylazine-induced anesthesia.

### 2.2. Ad7 Virus and DNA Purification

Adenovirus type 7 Gomen strain (ATCC VR-7) was purchased from ATCC. The Ad7 Gomen strain was obtained from a throat swab of a California military recruit with pharyngitis in 1954 [18]. The strain used to vaccinate military recruits is Ad7a; however, the Ad7 prototype Gomen strain was used in this study due to commercial availability. Final amplification of the virus was performed in a Corning 10-cell stack (~6300 cm^2^). Virus was amplified in 293 cells and purified on two sequential CsCl ultracentrifuge gradients. Viral genomic DNA (gDNA) was purified using high-titer Ad7 virus and the PureLink Viral RNA/DNA mini kit (Invitrogen, Carlsbad, CA, USA).

### 2.3. Development of an Ad7 Fiber Expressing Cell Line

The Ad7 fiber sequence was codon-optimized (co) and synthesized by Genscript. Codon-optimization will improve protein expression levels and significantly reduce the possibility of homologous recombination in vitro. The Ad7-Fiber-co was cloned into pCT-IRES-hrGFP and transfected into 293 cells using Polyfect (Qiagen, Valencia, CA, USA). Twenty-four hours after transfection, Geneticin was added (0.5 mg/mL). The cell line was passaged with Geneticin for several weeks before Western blot analysis of Ad7 fiber expression and cryopreservation [19] (Appendix A).

### 2.4. Recombinant Ad7 Plasmid Construction

The complete Ad7 genome was cloned into a single low copy plasmid, and the E1 and E3 deleted viruses expressing eGFPLuc were created as previously described [20] (Figure 1). Table 1A shows the primers used to create the shuttle plasmid that was used to replace the fiber gene. The entire fiber protein, from position 31320–32297 of the Ad7 genome, was deleted and replaced by a DsRed reporter gene. The shuttle was created using an overlapping PCR product that contained a unique AscI restriction enzyme site. A schematic representation of the cloning PCR fragment is shown (Appendix A). DsRed was fused to LoxP-Blasticidin-LoxP by overlapping PCR. Table 1B shows the primers used to create the final PCR products (Appendix A). AscI restriction sites were incorporated into the PCR primers for ligation into the shuttle plasmids. Both PCR amplified transgenes were cloned into TOPO pCR8 (Invitrogen, Carlsbad, CA, USA) and confirmed by sequencing. A final schematic representation of the fragment used for homologous recombination is shown (Appendix A).

To make the recombinant single cycle Adenovirus 7 (scAd7) plasmid, the shuttle plasmid was digested with PmeI. The digested shuttle plasmid was co-transformed with 1.0 µg of plasmid containing the complete Ad7 genome (pAd7) into BJ5183 electrocompetent cells. Recombinants were selected on LB containing Kanamycin and Blasticidin and then confirmed by restriction digestion analysis. Positive recombinant clones were transformed into XL-1 cells and maxiprepped using the Qiagen HI Speed Maxi kits (Qiagen, Valencia, CA, USA).

### 2.5. Recombinant Ad7 Rescue and Purification

The recombinant Ad7 genomes were released from the plasmid backbone by digestion with AsiSI. The digested plasmids were buffer-exchanged using a StrataPrep PCR purification kit (Agilent Technologies, Santa Clara, CA, USA). All transfections were performed in 6-well plates as described by Polyfect Transfection reagent (Qiagen, Valencia, CA, USA). The linearized Ad7-ΔE1 and Ad7-ΔE3 gDNA was transfected into 293 cells, and scAd7 gDNA was transfected into 293-Ad7-Fiber cells (Figure 1). GFP and DsRed expression were used to track foci as the viruses began to rescue. Viruses were released from infected cells by 3 freeze–thaw cycles and amplified in sequential passages in the complementing host cell lines. The viruses were amplified and purified as previously described for wt Ad7. The virus particle quantity was determined on a NanoDrop Lite spectrophotometer (Thermo Fisher) with an OD260. Ad7-ΔE1, Ad7-ΔE3, and scAd7 produced 4095, 10,600, and 14,400 vp/cell, respectively.

### 2.6. In Vitro Reporter Gene Expression and Luciferase Assay

The indicated cell lines were grown in 6-well tissue culture treated plates and infected with 50 vp/cell (Corning, Corning, NY, USA). Images were taken at 24 and 72 h post-infection (Figure 2). The luciferase activity of the viruses was measured by infecting (50 vp/cell) 293, 293-Ad7-Fiber, and A549 cells grown in 96-well black assay plates (Corning, Corning, NY, USA). At 24 h time points, the cells were lysed with 5× passive lysis buffer and luciferase activity was detected by adding 50 μL of luciferase activity reagent (LAR) (Promega, Madison, WI, USA). The plate was shaken on an orbital shaker, and luciferase activity was measured using a Beckman Coulter DTX 880 Multimode Detector (Figure 2).

### 2.7. In Vitro qPCR of Viruses

Quantitative PCR (qPCR) was performed using a ViiA7 Sequence detection system (Applied Biosystems, Warrington, UK). Table 1C shows the primers used for the detection of human β-Actin (ACTB), hexon, fiber, GFPLuc, and DsRed genes. Both 293 and A549 cells were grown in 6-well tissue culture plates and infected with the indicated virus at 50 vp/cell in 5% FBS DMEM. Total DNA was isolated from the cells at 24 and 72 h post-infection using the DNAeasy Blood and Tissue Kit (Qiagen Valencia, CA, USA). Standard curves were made for each quantitated gene, and ACTB was used as a standard to determine gene copies/cell. Real-time qPCR was done using Power SYBR Green PCR Master Mix, 100 ng of purified total DNA, 3.0 μM primers, and the following reaction conditions: 95 °C for 15 s, 56 °C for 15 s, and 72 C for 15 s (40 cycles).

### 2.8. Western Blot

To compare Ad7 hexon expression between the developed Ad7 viruses, confluent 293 cells were infected with 500 vp/cell of the indicated virus. Cells were incubated at 37 °C and 5% CO_2_ and harvested at 48 h. Cells were denatured using Laemmli buffer plus 2 mercaptoethanol and boiled at 100 °C for 10 min. The sample was then passed through a QIAshredder (Qiagen), loaded onto a 12.5% SDS-PAGE gel, and separated using electrophoresis. Protein was transferred to a nitrocellulose membrane and blocked for 30 min with 5% milk in TBST. Primary antibody was sera from a Balb/c mouse infected with 10^10^ vp of wtAd7 virus through the intramuscular route and bleed 2 weeks later. The membrane was incubated in primary antibody at 1:200 and mouse anti-GAPDH (sc-47724; Santa Cruz Biotechnology, Inc., Dallas, TX, USA) at 1:2000 in TBST 1% milk overnight at 4 °C. After 3 washes in TBST, the membrane was incubated with goat anti-mouse-HRP conjugated antibody (Millipore Sigma, Burlington, MA, USA) at 1:2000 in TBST 1% milk for 1 h at room temperature (RT). After incubation, the membrane was washed and developed with SuperSignal West Pico Chemiluminescent Substrate (Thermo Scientific, Waltham, MA, USA).

### 2.9. Virus Progeny Assay

To examine the virus progeny produced after infection of noncomplementing cells lines, A549 cells were infected with each virus at an MOI of 1. MOI was determined by FACS on serial dilutions of each virus to determine the FACS infectious units (IFU)/mL (Appendix A). The A549 cells were infected for 1 h, washed twice with PBS, fresh DMEM 10% FBS was added, and they were incubated at 37°C with 5% CO_2_. Cells were harvested at 48 h by spinning down the cells, discarding supernatant, and suspending cells in 300 uL Ad-tris buffer. Next, three consecutive freeze–thaws were performed to release the virus progeny, and all of this supernatant was plated on fresh A549 cells. Cells were harvested for flow cytometry at 48 h and 7 days post-infection with the freeze/thawed supernatant. Cells were washed, fixed in 1% formalin, and then analyzed for GFP expression on a Cytek DxP10 flow cytometer. Infections were performed in triplicate.

### 2.10. Anti-Ad7 Neutralization Titer

Human desmoglein (hDSG) transgenic mice were immunized intramuscularly with 10^10^ vp per mouse of Ad7-ΔE1, Ad7-ΔE3, or scAd7Adenovirus vectored vaccination, which has been shown to induce rapid and strong humoral and cellular immune responses [21,22]. Therefore, two weeks post-immunization, the mice were sacrificed, and sera was tested for anti-Ad7 neutralizing antibodies (NAbs). Sera was heat inactivated at 56 °C for 30 min before a serial 2-fold dilution was performed in a 96-well black plate (3603 Corning). Naïve mouse serum was used as negative controls. Fifty microliters of Ad7-dE3 expressing GFP-Luciferase (1 × 10^8^ Ad-GL vp/mL) was then added to the sera dilutions and incubated at 37 °C for 1 h. Next, 50 µl of A549 cells at 1 × 10^5^ cells/mL was added to each well, and the plates were incubated for 20 h at 37 °C and 5% CO_2_ before readout. The luciferase activity was determined as described above. Neutralizing antibodies titers were determined as the reciprocal dilution to inhibit 50% of luciferase activity as compared to no serum controls.

### 2.11. Elipsot

Ninety-six-well polyvinylidene difluoride-backed plates (MultiScreen-IP, Millipore, Billerica, MA) were coated with 50 μL of anti-mouse IFN-γ mAb AN18 (5 mg/mL; Mabtech, Stockholm, Sweden) overnight at 4 °C. The plates were blocked with cRPMI medium at 37 °C for 1 h. Splenocytes were isolated from mice using a 70 μm Nylon cell strainer (BD Labware, Franklin Lakes, NJ, USA), red blood cells were lysed using ACK lysis buffer, and the splenocytes were resuspended at a concentration of 5 × 10^6^ splenocytes/mL. Human DSG transgenic mice are on a mixed genetic background so antigen-presenting cells were created. Naïve hDSG splenocytes were isolated and infected with an Ad5-∆E1/3 replication-defective virus expressing GFP-Luciferase at 5000 vp/cell and incubated overnight at 37 °C with 5% CO_2_. Single-cell suspensions of splenocytes from vaccinated mice were incubated 1:1 with APCs in duplicate and incubated overnight at 37 °C with 5% CO_2_. The plates were washed 6× with PBS and incubated with 100 μL of biotinylated anti-mouse IFN-γ mAb (1:1000 dilution; Mabtech) diluted in PBS with 1.0% FBS for 1 h at RT. Plates were washed 6× with PBS and incubated with 100 μL of streptavidin–alkaline phosphatase conjugate (1:1000 dilution; Mabtech, Cincinnati, OH, USA) diluted in PBS 1.0% FBS. After 1 h at RT, the plates were washed 6× with PBS. To develop, 100 μL of BCIP/NBT (Plus) alkaline phosphatase substrate (Thermo Fisher) was added to each well, and development was stopped by washing several times in dH2O. The plates were air-dried and spots were counted using an automated ELISpot plate reader (AID iSpot Reader Spectrum, Oceanside, CA, USA). Results are expressed as spot-forming cells (SFC) per 10^6^ splenocytes.

## 3. Results

### 3.1. Development of a Single-Cycle Ad7 Virus

We developed a single-cycle Ad7 virus (scAd7), along with replication-competent and replication-defective Ad7 viruses, in order to examine the scAd7 virus’s potential as a vaccine candidate. Reporter genes were inserted to evaluate these viruses using in vitro studies. The replication-competent Ad7 virus (Ad7-∆E3) has an eGFP-Luciferase (GFP-Luc) fusion expression cassette running off a CMV promoter in place of the nonessential Ad E3 gene (Figure 1A) and can produce infectious virus without any additional genes. The replication-defective Ad7 virus (Ad7-∆E1) has the essential E1 gene deleted and relies on E1 to be supplied by a complementing 293 cell line in order to replicate (Figure 1B). Since E1 is required for replication, this virus will not replicate its genome or produce any virus outside of the complementing 293 cell line. The single-cycle Ad7 (scAd7) has a deletion of both the E3 gene and the fiber gene, which is an essential structural gene supplied in trans by a complementing 293-Ad7-Fiber cell line (Figure 1C). When scAd7 is grown outside of the 293 Ad7-fiber cell line, it only produces fiberless virus, which has severely reduced infectivity [15,16]. Therefore, the scAd7 virus will replicate its genome ≥10,000-fold and express Ad7 viral components but will only assemble fiberless particles that have impaired infectivity (Figure 1C). We used DsRed as the reporter gene replacing the Ad fiber gene, which runs off the Ad major late promoter (MLP) during replication. This scAd7 virus will express Ad7 genes, GFP, luciferase, and DsRed transgenes in human cells.

### 3.2. In Vitro GFP Expression and Luciferase Activity

To examine reporter gene expression in complementing and noncomplementing cells lines, 293, 293-Ad7-Fiber, and A549 cells were infected with all three Ad7 viruses at 50 vp/cell. Images were taken at 24 and 72 h post-infection. The scAd7 virus showed equivalent GFP expression in all three cell lines. The replication-defective Ad7-ΔE1 virus showed the best GFP expression in the E1 complementing 293 cells, whereas the replication-competent Ad7-ΔE3 virus showed the greatest levels of expression in A549 cells (Figure 2A). This was further supported by evaluating the luciferase activity of the three Ad7 viruses at different time points (Figure 2B–D). As with the florescence expression, the Ad7-ΔE1 virus showed the strongest luciferase activity in 293 cells, while the Ad7-ΔE3 virus showed the strongest activity in A549 cells. Peak luciferase activity was observed at 3 days post-infection for all viruses.

### 3.3. Ad7 Hexon Expression and gDNA Replication In Vitro

The deletion of the Ad7 fiber gene and insertion of the GFP-Luciferase and DsRed transgenes were confirmed by PCR amplification of virus genomic DNA (Figure 3A). No Ad7 fiber gene was detected in the viral genomic DNA (gDNA) of the scAd7 virus confirming successful deletion of this gene and replacement with DsRed. 

In contrast to the Ad7-∆E1 replication-deficient virus, the scAd7 virus can express Ad7 protein products in noncomplementing cells and therefore potentially increase the anti-Ad7 immune response. To confirm strong Ad7 protein expression in the scAd7 virus, a Western blot for Ad7 hexon was performed (Figure 3B). Noncomplementing A549 cells were infected with the indicated virus at 500vp/cell, washed, incubated for 48 h, and cell lysates were collected. No Ad7 hexon expression was detected in the Ad7-∆E1 infection because this virus is not able to replicate its genome or produce Ad7 proteins outside of E1 complementing cells. However, the scAd7 virus had strong hexon expression, which was comparable to that of the replication-competent Ad7-∆E3 virus.

To further evaluate the scAd7 virus’s ability to replicate its gDNA, we infected complementing 293 cells and noncomplementing A549 cells with the three viruses and determined the relative genome copies at 24 and 72 h post-infection by real-time quantitative PCR (qPCR) (Table 1). As expected, all three Ad7 viruses were able to replicate their gDNA in 293 cells (Figure 3C). However, in the A549 cells, the replication-defective Ad7-ΔE1 was not able to replicate its gDNA, whereas the scAd7 virus was able to replicate its gDNA in A549 cells which do not have a complementing fiber gene (Figure 3D). The scAd7 virus had similar gDNA replication kinetics as compared to the replication-competent Ad7-ΔE3 virus in the A549 cells.

### 3.4. scAd7 Virus Infection Produces Virus Progeny with Impaired Infectivity

To confirm that infection by the scAd7 virus produces virus progeny with impaired infectivity, noncomplementing A549 cells were infected at an MOI of 1 with the indicated virus for 1 h, washed, and incubated for 48 h. This initial infection showed comparable numbers of GFP+ cells (Figure 4A,B). These infected cells were harvested at 48 h, freeze/thawed to release the virus progeny, and then all of the supernatant was used to infect new A549 cells. GFP+ cells were then quantified at 48 h and 7 days post-infection with the freeze/thawed supernatant (Figure 4A,B). The replication-competent Ad7-∆E3 showed significant production of infectious virus particles in the noncomplementing cell line, while the replication-deficient Ad7-∆E1 did not. The scAd7 virus showed severely reduced infectivity of virus progeny (Figure 4A,B). There is limited transduction of the fiberless virus due to interactions between the penton base and cellular αvβ integrins; however, there is no amplification and spread of the fiberless virus outside of the fiber-complementing cell line (Figure 4B).

### 3.5. In Vivo Immunogenicity of the Ad7 Vaccine Constructs

Human desmoglein 2 (hDSG) was determined to be the cellular receptor for Ad7 [23]. Human DSG transgenic mice were immunized via the i.m. route with 10^10^ vp/mouse of the indicated virus and sacrificed 2 weeks later. An anti-Ad7 neutralization assay was performed to evaluate neutralizing antibodies (Figure 5A). All three Ad7 viruses showed significant anti-Ad7 antibodies. However, Adenovirus does not replicate in mice [24], so differences between the three viruses were not observed. In addition, we observed a cellular immune response against the reporter gene luciferase in all three viruses (Figure 5B). 

## 4. Discussion

Adenovirus type 7 infection is associated with acute respiratory disease. However, the vaccine to protect against Ad7 infections is only licensed for use in the military and uses wild type virus which can be shed in the stool for up to 28 days after immunization. Here we developed a single-cycle Ad7 virus as a safer Ad7 vaccine for use in the general public. The single-cycle virus is deleted for the fiber protein and therefore assembles fiberless Ad progeny during infection of noncomplementing cells. Interaction between fiber and the cellular protein hDSG is the primary mode of transduction for Ad7 [23]; therefore, this fiberless virion has severely reduced infectivity. 

Since we do not have a cell line that complements the Ad7 E1 genes, we used 293 cells that express complementing Ad5 E1 genes. We were able to produce our Ad7-ΔE1 virus; however, the resulting virus had a high virus particle to IFU ratio (Appendix A). This is not unusual since E1 genes from different species of Adenovirus do not always fully complement each other and have to be grown in E1 and E4 complementing cell lines [25]. Establishing a cell line that expresses the species B E1 genes may significantly improve the viral particle to IFU ratio. 

This scAd7 virus shows strong Ad7 hexon expression (Figure 3B) and strong gDNA replication (Figure 3D), even in noncomplementing A549 cell lines. These levels are comparable to that of the Ad7-∆E3 replication-competent virus. By contrast, the Ad7-dE1 virus showed undetectable hexon expression after infection of noncomplementing A549 cells. While we did not detect any late gene synthesis after infection with our E1 deleted virus, studies using a high MOI infection (>500) of A549 cells detected low late gene expression; however, it was not indicative of efficient virus replication [26].

Our in vitro studies support that the virus progeny produced from a scAd7 infection of noncomplementing cells has severely reduced infectivity (Figure 4). The limited transduction seen with the scAd7 virus in the virus progeny assay could be attributed to minor interacts between the penton base and cellular αvβ integrins [15]. In vivo studies showed development of strong anti-Ad7 neutralizing antibodies by all Ad7 viruses (Figure 5A). However, differences between the Ad7 virus constructs could not be observed due to the lack of permissive Adenovirus replication in mice [24]. In addition, there is variation in hDSG receptor expression in different tissues of the transgenic mice [17]. While intranasal or enteric route of immunization might increase transgene expression in these mice, replication of Adenovirus would still not occur. A small animal model that was transgenic for hDSG and permissive for human Adenovirus replication would greatly improve our ability to evaluate the scAd7 vaccine platform. Unfortunately, there is currently no available small animal model to optimally support preclinical studies.

The scAd7 virus currently has reporter transgenes; however, future work could examine substituting these genes for pathogen genes, such as the influenza virus hemagglutinin, in order to create a multivalent vaccine vector. In fact, a single-cycle construct based on Ad6 has already shown promise at inducing anti-influenza and anti-Ebola virus immunity [27,28]. By replacing the GFP, luciferase, and DsRed genes, it could be possible to create a single-cycle Ad7 virus that has utility against itself, and three other genes that could easily represent three distinct pathogen genes. In addition, this single-cycle approach could be applied to other medically important Ad types, such as Ad4, which is also associated with ARD. 

Our scAd7 virus combines the safety advantages of a replication-defective virus with the increased Ad7 protein expression of a replication-competent virus. Virus progeny produced outside of the complementing cell line have impaired infectivity. However, the scAd7 virus is potentially more immunogenic than a replication-defective virus because it retains the ability to replicate its gDNA, thus increasing Ad7 protein expression. Due to these advantages, we believe that scAd7 viruses should be further studied as an alternative, safer Adenovirus 7 vaccine.

## Figures and Tables

**Figure 1 viruses-11-00413-f001:**
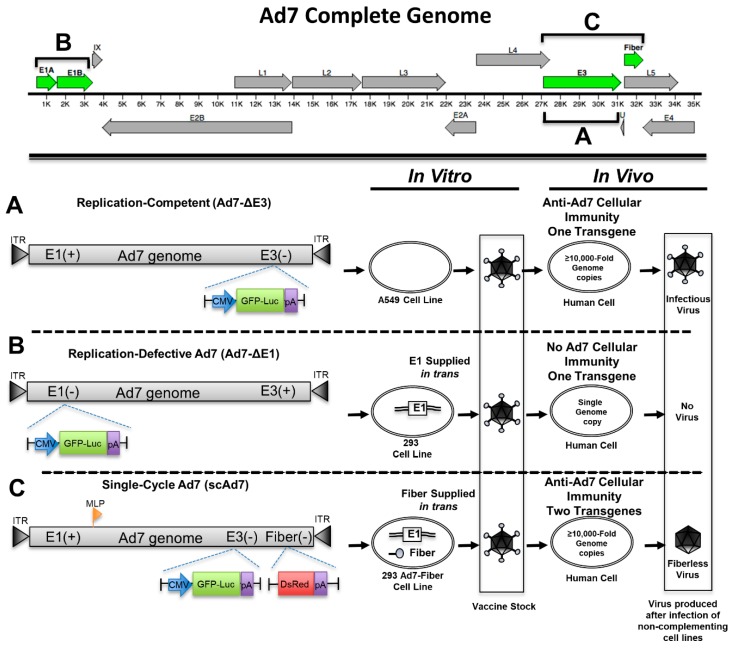
Genomic organization of three Adenovirus type 7 (Ad7) viruses. The complete Ad7 genome can be seen at the top with the manipulated Adenovirus genes highlighted in green. The replication-competent virus platform has the nonessential E3 genes replaced with a GFP-Luciferase (GFP-Luc) expression cassette (**A**). The replication-defective Ad7 virus has the essential E1 genes deleted and replaced with a GFP-Luciferase expression cassette. This virus requires the E1 gene in trans from the 293 cells in order to replicate (**B**). The single cycle Ad7 (scAd7) virus has the E3 genes and fiber deleted (**C**). The E3 genes are replaced with a GFP-Luciferase expression cassette, and the fiber gene has been replaced with a DsRed gene running off the major late promoter (MLP) during replication. The scAd7 virus can only produce an infectious virus in Ad7 fiber complementing cell line. However, transduction of normal cells results in ≥10,000-fold genome amplification and only produces fiberless virus with severely impaired infectivity.

**Figure 2 viruses-11-00413-f002:**
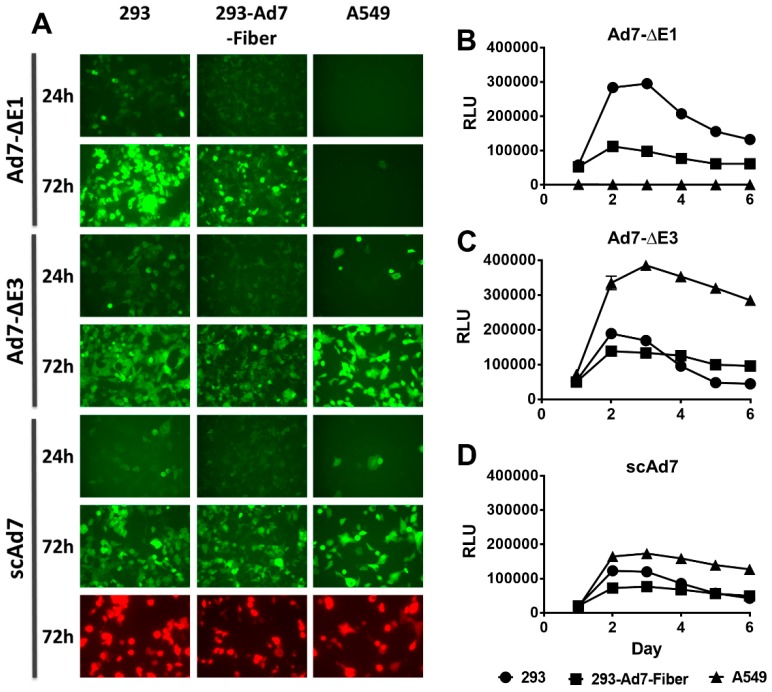
In vitro reporter gene expression of the Ad7 viruses in three cell lines 293, 293-Ad7-Fiber, and A549 cells, grown in 6-well tissue culture plates and infected with the indicated viruses at 50 vp/cell (**A**). Fluorescent images of GFP expression were taken at 24 and 72 h post-infection at 10× magnification. All pictures were imaged at the same light intensity for comparison of reporter gene expression. The luciferase kinetics of each virus was also examined by infecting 293, 293-Ad7-Fiber, and A549 cells with the indicated viruses at 50 vp/cell. Luciferase expression was determined at 24 h intervals for six days. The luciferase expression kinetics of the Ad7-ΔE1, Ad7-ΔE3, and scAd7 are shown in panels B, C, and D, respectively. Data are expressed as relative luminescence units (RLU) and error bars indicate standard error for 4 plates replicates.

**Figure 3 viruses-11-00413-f003:**
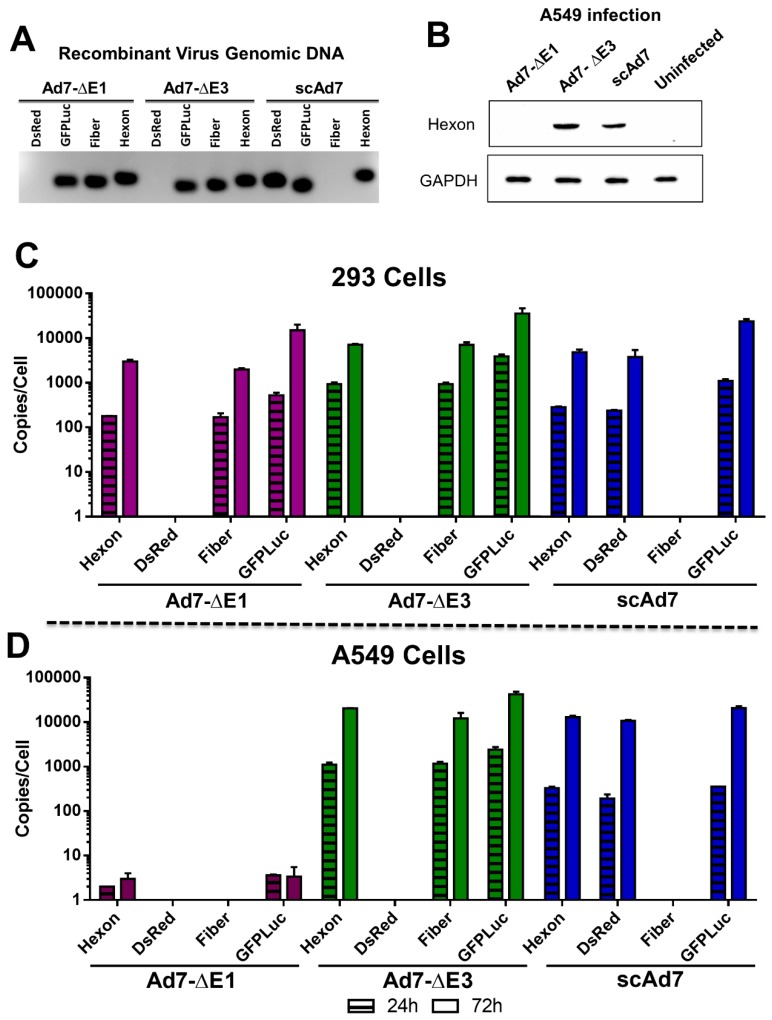
In vitro Ad7 hexon expression and gDNA replication. The absence of the fiber gene and the presence of the DsRed transgene in the scAd7 viral genomic DNA was confirmed by PCR (**A**). Hexon protein expression in noncomplementing cells was confirmed by Western blot of A549 cells infected with the indicated virus at 500 vp/cell and harvested after 48 h. GAPDH is used as a cellular control (**B**). In addition, we quantitated the gDNA replication kinetics of all three Ad7 viruses in E1 complementing 293 cells (**C**) and noncomplementing A549 cells (**D**). This was achieved by quantifying the introduced genes, DsRed and GFPLuc, and the structural genes, fiber and hexon. Gene copies are displayed per cell as determined by the ACTB gene. Real-time qPCR was performed in triplicate, and error bars indicate SEM.

**Figure 4 viruses-11-00413-f004:**
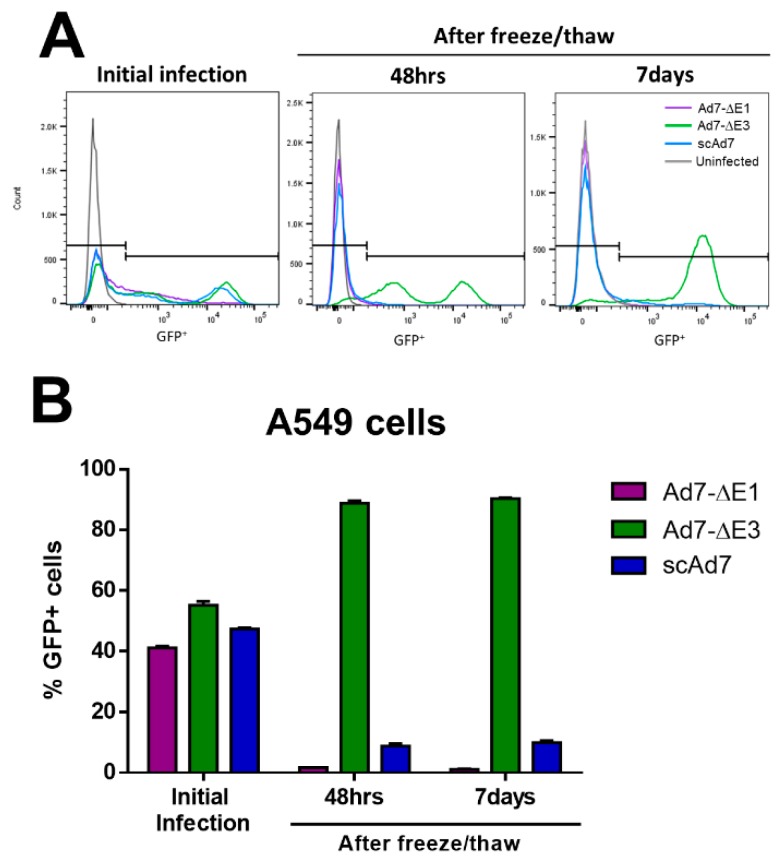
Virus progeny of the three Ad7 viruses after infection of noncomplementing A549 cell lines. Noncomplementing A549 cells were infected at an MOI of 1 with the indicated virus. At 48 h after this initial infection, flow cytometry was performed to examine GFP+ cells (**A**) and quantitate the percent GFP+ cells (**B**). A duplicate well of infected cells was harvested, freeze/thawed three times to release virus progeny, and the supernatant was added to new A549 cells. After addition of the freeze/thaw supernatant, cells were harvested for flow cytometry 48 h and 7 days after infection to examine the presence of infectious virus progeny. Infection was performed in triplicate, and error bars represent SEM.

**Figure 5 viruses-11-00413-f005:**
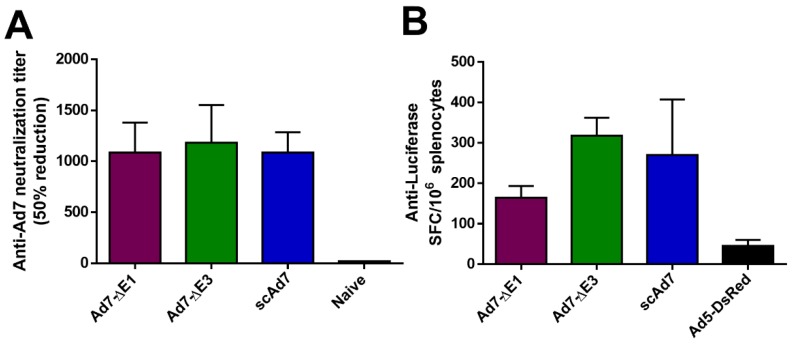
In vivo Ad7 neutralization and Anti-Luciferase cellular immune response in hDSG transgenic mice. Human desmoglein (hDSG) transgenic mice (*n* = 5) were vaccinated via the i.m. route with 10^10^ vp per mouse of the indicated virus. The anti-Ad7 antibody response was determine by a neutralization assay with 50% reduction of luciferase expression (**A**). The cellular immune response against the luciferase reporter was examined with an ELISPOT (**B**). Mice were immunized with a replication-deficient Ad5 vector expressing only DsRed as a negative control in this luciferase ELISPOT. Data are expressed as the mean with standard error (SEM).

**Table 1 viruses-11-00413-t001:** Primers for the creation and testing of the scAd7 vaccine. These primers were used for the PCR amplification of the homologous flanking DNA used for inserting the DsRed gene in place of the fiber gene (**A**). Ad7-Fib-Ins-F and Ad7-Fib-Ins OL-R primer pairs were used to amplify the 500 bp region 5′ of the fiber gene. Ad7-Fib-Ins-OL-F and Ad7-Fib-Ins-R primer pairs were used to amplify the 500 bp region 3″ of the fiber gene. The two PCR products were combined, and the Ad7-Fib-Ins-F and Ad7-Fib-Ins-R primers were used to amplify the overlapping PCR products. These primers were used to create the DsRed floxed blasticidin gene for incorporation into the shuttle plasmid (**B**). The overlapping PCR products were amplified similar to the previously described the shuttle PCR product. The following PCR primer sets were used to quantitate the viral genes, cellular Actin gene, and the introduced DsRed and GFP-Luc genes (**C**).

**A. Primers for Ad7 Fiber Deletion/Gene Replacement Shuttle Plasmids.**			
Ad7-Fib-Ins-F	GTTTAAACGAACGCGTGACCAAAGAGCTCAGAG		
Ad7-Fib-Ins-OL-R	AATAAACAAGTTAAACTTTATTTTGTGGCGCGCCCTGGGAAGAAAGACATGAAGATTGTG		
Ad7-Fib-Ins-OL-F	CACAATCTTCATGTCTTTCTTCCCAGGGCGCGCCACAAAATAAAGTTTAACTTGTTTATT		
Ad7-Fib-Ins-R	GTTTAAACTAATCTAAGTGAAATCAGAATGCGT		
**B. Primers for DsRed.**			
AscI-DsRed-F	GGCGCGCCATGGCCTCCTCCGAGAACGTCATCA		
DsRed-LBL-R	AAGGGCGAATTCGGAGCCTGCTTTTTTCTACAGGAACAGGTGGTGGCGGCCCTCGGTG		
DsRed-LBL-F	CACCGAGGGCCGCCACCACCTGTTCCTGTAGAAAAAAGCAGGCTCCGAATTCGCCCTT		
AscI-LBL-R	GGCGCGCCCAAGAAAGCTGGGTCGAATTCGCCC		
**C. qPCR Primers**			
DsRed-F	CACTACCTGGTGGAGTTCAAG	DsRed-R	GATGGTGTAGTCCTCGTTGTG
Ad7 Fiber-F	GATTCCTTCAACCCTGTCTACC	Ad7 Fiber-R	CCGTCTGGGCTTTGTGTAA
Ad7 Hexon-F	GGAACCTTACCCAGCCAATTA	Ad7 Hexon-R	AGTTGCTGGAGAAGGGAATG
ACTB gDNA-F	GGCCTTGGAGTGTGTATTAAGT	ACTB gDNA-R	GGACATGCAGAAAGTGCAAAG
eGFPLuc-F	CGGAAAGACGATGACGGAAA	eGFPLuc-R	CGGTACTTCGTCCACAAACA

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
