# Peer review of "A Single-Cycle Adenovirus Type 7 Vaccine for Prevention of Acute Respiratory Disease"

_viruses, 2019, doi:10.3390/v11050413_

Round 1

Reviewer 1 Report

The revised version of manuscript ID: Viruses-439370 has improved forward. Some text and sentences were changed and become better. Moreover, the Table S2 data demonstrated that the fiber defective vectors is reduced infectivity but can replicate in outside of complement cell lines in  comparison to Ad7 ΔE1 virus.

My two comments:
Authors did not response to my previous comment Q3 as followings: Line 250-251, Fig.3A and 3B: Ad7-ΔE1 hexon was positive in PCR assay but negative in WB. The data is very difficultly to understand why the Ad7 virus with E1 deletion results in the hexon negative.  The hexon expression is mainly controlled by major late promoter that exists in the Ad7- ΔE1 viral genome. Therefore, the data is not true.

The title of the manuscript with “A single-cycle” adenovirus type 7 vaccine for prevention of acute respiratory disease. The “a single-cycle” are incorrect to describe Ad7-ΔFib virus and should be changed, due to that the Ad7-Δfib is able to infect and replicate in a target cell line via the secondary receptors: ανβ3 or ανβ5.

Author Response

Response to Reviewers

Thank you very much for your time and efforts to improve our manuscript.  We have addressed all of your comments.  The critiques and comments are underlined and our responses are italicized.  We hope you find our revisions suitable.

Reviewer #1

The revised version of manuscript ID: Viruses-439370 has improved forward. Some text and sentences were changed and become better. Moreover, the Table S2 data demonstrated that the fiber defective vectors is reduced infectivity but can replicate in outside of complement cell lines in comparison to Ad7 ΔE1 virus. 

My two comments: 
1) Authors did not response to my previous comment Q3 as followings: Line 250-251, Fig.3A and 3B: Ad7-ΔE1 hexon was positive in PCR assay but negative in WB. The data is very difficultly to understand why the Ad7 virus with E1 deletion results in the hexon negative.  The hexon expression is mainly controlled by major late promoter that exists in the Ad7- ΔE1 viral genome. Therefore, the data is not true.

293 cells express the Adenovirus E1 gene products and are used to grow E1 deleted replication-defective viruses.  A549 cells do not express the Adenovirus E1 gene products and can only be used to grow replication-competent, or wildtype, viruses.  In the absence of the E1 gene products, there is no transcriptional activation of any viral genes.  The transgenes in the recombinant viruses are under the control of the immediate/early cytomegalovirus (CMV) promoter are not controlled by the E1 gene products. Therefore, the transgenes that are under the control of the CMV promoter are expressed in both the 293 and A549 cells. In order to reduce confusion, we have included the following statement in the results section for Figure 3:

“Transcriptional control of the Adenovirus genes is regulated by the E1 gene products.  Upregulation of the late genes is controlled by E1 and E4 gene products in combination. Therefore E1 is the “gatekeeper” for all Aenovirus gene regulation.  Since A549 cells do not express the Adenovirus E1 gene products, no viral genes are transcribed by the replication-defective Ad7-∆E1 virus. Therefore, no Ad7 hexon expression was detected in the Ad7-∆E1 infection because this virus is not able to replicate its genome or produce Ad7 proteins outside of E1 complementing cells.”

2) The title of the manuscript with “A single-cycle” adenovirus type 7 vaccine for prevention of acute respiratory disease. The “a single-cycle” are incorrect to describe Ad7-ΔFib virus and should be changed, due to that the Ad7-Δfib is able to infect and replicate in a target cell line via the secondary receptors: ανβ3 or ανβ5.

There may be some misunderstanding between replication and transduction.  We agree that there is a limited amount of transduction of the fiber less virus through the αvβ integrins interactions, but this is self-limiting and is not sufficient for viral amplification.  We have included the following statement in the results section:

There is limited transduction of the fiberless virus due to interacts between the penton base and cellular αvβ integrins, however there is no amplification and spread of the fiberless virus outside of the fiber-complementing cell line (Fig. 4B).”

We found that fiberless virus progeny have severely reduced infectivity because of lack of the high affinity fiber and hDSG interactions. Therefore, propagation of this virus is limited and we feel that single cycle is currently the best description of the unique strategy presented here.  

Reviewer 2 Report

ScAd7 vaccine.   This article is now in a better shape. 

Line 23 the previous statement " are not infectious has now been changed to   " severely reduced infectivity" which is a more careful formulation .Fig4B  demonstrates that  the progeny was definitely reduced  but some was produced after infection of noncomplementing A549 cells

 Line 85 Note that  the Enteric coated adenovirus vaccine used by military recruits since 1971 was based on  the Ad7a genometype  and not  the Ad7 prototype the strain Gomen.

Figure 2 the mixup of figures has now been amended Fig2B The fibers are provided by the 293-Ad7-fiber cell The yield of  Ad7 virionswith deleted E1

was conspicously low in these cells. Is this a consequence of an uncontrolled amount of fiber expressed in  these cells? See  Levine AJ Ginsberg HS J Virol Vol 1 no 4, 747-757  1967

The dominating neutralizing epitopes are expressed on the hexons. Still  it  would  be valuable to assess  the in vivo  protection against  infection  obtained  by immunization by fiberless adenovirus particles in a more  permissible animal model than the hDSG mice  

Minor  line 315  fluorescence  line 463  interaction

Author Response

Response to Reviewers

Thank you very much for your time and efforts to improve our manuscript.  We have addressed all of your comments.  The critiques and comments are underlined and our responses are italicized.  We hope you find our revisions suitable.

Reviewer #2

ScAd7 vaccine: This article is now in a better shape.

1) Line 23 the previous statement " are not infectious has now been changed to   " severely reduced infectivity" which is a more careful formulation . Fig4B demonstrates that  the progeny was definitely reduced  but some was produced after infection of noncomplementing A549 cells

We agree that there is a limited amount of transduction of the fiberless virus through the αvβ integrins interactions, but this is self-limiting and is not sufficient for viral amplification.  We have included the following statement in the results section:

There is limited transduction of the fiberless virus due to interacts between the penton base and cellular αvβ integrins, however there is no amplification and spread of the fiberless virus outside of the fiber-complementing cell line (Fig. 4B).”

2) Line 85 Note that  the Enteric coated adenovirus vaccine used by military recruits since 1971 was based on  the Ad7a genome type  and not the Ad7 prototype the strain Gomen.

We have added this to the methods to aid with clarification. 

3) Figure 2 the mixup of figures has now been amended Fig2B The fibers are provided by the 293-Ad7-fiber cell The yield of  Ad7 virions with deleted E1 was conspicuously low in these cells. Is this a consequence of an uncontrolled amount of fiber expressed in these cells? See  Levine AJ Ginsberg HS J Virol Vol 1 no 4, 747-757  1967

This is a possible explanation for the reduce expression seen in this cell line however the effect is not as dramatic in the Ad7-dE3 or the scAd7 so it is difficult to draw a conclusion that it is due to the cell line alone. 

In order to address this issue, the following statement is included in the Discussion: “We were able to produce our Ad7-ΔE1 virus, however the resulting virus had a high virus particle to IFU ratio (Table S2).  This is not unusual since E1 genes from different species of Adenovirus do not always fully complement each other and have to be grown in E1 and E4 complementing cell lines [26].  Establishing a cell line that expresses the species B E1 genes may significantly improve the viral particle to IFU ratio.”

4) The dominating neutralizing epitopes are expressed on the hexons. Still  it would  be valuable to assess  the in vivo protection against  infection  obtained by immunization by fiberless adenovirus particles in a more  permissible animal model than the hDSG mice  

We agree.  However, there are currently no small animal models that will both support Adenovirus replication and be transgenic for hDSG, which is needed for vaccination. The following sentences were added to the Discussion:

“A small animal model that was transgenic for hDSG and permissive for human Adenovirus replication would greatly improve our ability to evaluate the scAd7 vaccine platform.  Unfortunately, there is currently no available small animal model to optimally support preclinical studies.”

5) Minor  line 315  fluorescence line 463  interaction

Corrected

Round 2

Author Response

 Manusript ID viruses -489393. 

The authors reply do not sufficiently support their own results and is against previous publication by Imler JL., et. al., 1996. Gene Therepy). A549 cells, a cancer cell line, some oncogenes in the cells, play the role of E1A proteins, can initial the virus replication. Moreover, the major late promoter in Ad7ΔE1 genome can promote the hexon gene expression. Authors did not answer to my question correctly, since my question did not relate to the E1 expression cassette rather than the hexon, a capsid protein in Fig 3B., in which Hexon is negative for Ad7ΔE1 virus. Therefore, I suggest that Authors should repeat the experiments in Fig. 3B and run Western blot with qualified Ad7ΔE1 virus to identify the hexon expression, in order to match their own results in Fig 3A, where PCR hexon positive. 

Gene Ther. 1996 Jan;3(1):75-84. 

Novel complementation cell lines derived from human lung carcinoma A549 cells support the growth of E1-deleted adenovirus vectors.Imler JL1, Chartier C, Dreyer D, Dieterle A, Sainte-Marie M, Faure T, Pavirani A, Mehtali M. 

Author information 

Abstract 

Replication-defective E1-deleted adenoviruses are attractive vectors for gene therapy or live vaccines. However, manufacturing methods required for their pharmaceutical development are not optimized. For example, the generation of E1-deleted adenovirus vectors relies on the complementation functions present in 293 cells. However, 293 cells are prone to the generation of replication competent particles as a result of recombination events between the viral DNA and the integrated adenovirus sequences present in the cell line. We report here that human lung A549 cells transformed with constitutive or inducible E1-expression vectors support the replication of E1-deficient adenoviruses. E1A transcription was elevated in most of the cell lines, and E1A proteins were expressed at levels similar to those of 293 cells. However, the levels of expression of E1A did not correlate with the efficiencies of complementation of E1-deleted viruses in A549 clones, since some clones complemented replication in the absence of induction of E1A expression.In addition, complementation of E1-deficient adenoviruses did not require expression of the E1B 55-kDa protein. Although these cell lines contain the coding and cis-acting regulatory sequences of the structural protein IX gene, they are not able to complement viruses in which this gene has been deleted. In contrast to 293 cells, such new complementation cell lines do not contain the left end of the adenoviral genome and thus represent a significant improvement over the currently used 293 cells, in which a single recombination event is sufficient to yield replication competent adenovirus.

Response:

The A549 cells referenced above have been transformed with the E1 region of Adenovirus. Although they construct A459 cells with inducible E1, they state that “most of these clones showed a clear morphological cell transformation similar to that of A549 clones transfected with the constitutive E1 expression vector (Figure 2), indicating that E1 expression occurred even in the absence of induction.”They also state that after infection with E1-deleted virus at an MOI of 2, “as expected, no amplification could be detected using untransfected A549 cells.” Therefore, this paper supports that E1 is necessary for Adenovirus replication. 

Fields virology states “Once the large E1A protein is expressed, it activates transcription of early regions E1B, E2 early, E3, and E4.” Thus, the E1 region of Adenovirus is the beginning of the transcriptional cascade needed for Adenovirus replication. Although our PCR shows that the hexon gene is present in the dE1 virus, this protein will not be expressed (as indicated by the Western Blot) because the major late promoter will not be activated without E1 present. 

More recent papers, such as Bratati, et al. 2017 PLoS One, have directly explored the Adenovirus gene expression in A549 cells and find the same result as our work. They perform an infection of A549 cells with E1 deleted Adenovirus and examine the expression of fiber by Western blot (Fig 5B-C). They find that fiber is detectable only at a very high MOI of 500. Fiber and hexon are both structural proteins so this data supports our findings as well. Our A549 infection was 500 virus particles (vp)/cell, making our MOI less than 2 for all of the virus constructs. This study concludes “Taken together, our work underscores the crucial role of E1A in orchestrating and modulating various cellular and viral processes to achieve efficient Ad replication. However, the results also clearly show that MOI-dependent effects on replication of E1-deleted Ad is an important consideration when using Ad vectors for gene delivery.”

To address this, we have added to the discussion: “In contrast, the Ad7-dE1 virus showed undetectable hexon expression after infection of non-complementing A549 cells. While we did not detect any late gene synthesis after infection with our E1 deleted virus, studies using a high MOI infection (>500) of A549 cells detected low late gene expression, however it was not indicative of efficient virus replication [27].”

This manuscript is a resubmission of an earlier submission. The following is a list of the peer review reports and author responses from that submission.

Round 1

Reviewer 1 Report

The manuscript ID: Viruses-439370  reported by Bullard et al. on “A single–Cycle Adenovirus type 7 vaccine for prevention of acute respiratory disease”. The authors described three constructs of adenovirus 7  viruses for vaccine purpose. The gene expression mediated by the three vectors were compared in 293, 293-7fiber, A549 cells, then progeny viruses from those vectors were further evaluated in A549 cells. Furthermore, immunoresponse to the three viruses were also investigated.

My comments:

Q1, Line 133, Fig.2:  Ad7-ΔE1 was not detectable in A549 cells. The photos for GFP expression and luciferase data are not sufficiently to demonstrate no infectious virus.   Quantity of each expression should compare in more precise methods: FACS assay, so that indicate the percent of  GFP positive  cells at 24 and 72 hr p.i. The results of Ad7-ΔE1 was negative in A549 cells were against early publication by Stone and Lieber (2005):  Stone et al demonstrated that the defective Ad11GFP can internalize and express GFP in A549 cells. Ad7 and Ad11 are very closed two viruses in species B adenoviruses. Authors should detect Ad7-ΔE1 virus particle under EM before to run the experiments.

Q2. Line 133, Fig.2A: Two RFP photos for scAd7 in 293 and 293-Ad7 fiber at 72hr p.i. are identical and why? Wrong placed photos?

Q3, Line 250-251, Fig.3A and 3B: Ad7-ΔE1 hexon was positive in PCR assay but negative in WB. The data is very difficultly to understand why the Ad7 virus with E1 deletion results in the hexon negative. The data is not true.

Q4, Line 133: Fig. 2A 2B versus Line 272: Fig.4A and 4B:  In non-competent A549 cells, GFP or luciferase expression mediated by Ad7-ΔE1 were negative in Fig2A,2B, but positive in Fig 4 A and 4B. Two results are contradictory.

Q5. Line 261-281: The experimental design for determination of infectious particles insufficiently demonstrated there is or not infectious virus particles. To ascertain that, end point titration and the toxicity experiment with loading serial diluted viruses and with a long observation periods (7-10 days) in A549 cells should be performed in order to know if there are infectious virus or not, or  reduced infectious ability for Ad7 vectors studied.

Q5. Line 285, Mice were immunized via…. and sacrificed 2 weeks later. The mouse experimental plan is not correct: due to a longer immunization period required for inducing neutralization antibody, the two weeks is not sufficient. Whereas to detect the cytokines: IFN-gamma a shorter period, just a couple of days can be detectable.  Therefore, authors should provide references to support their experiment plan is suitable.

Q6 Fig.5A: Positive control for Neutralization antibody is needed. Fig. 5B. Unclear function for Ad5-DsRed in anti-luciferase detection. If Ad5-DeRed also carrying Luc genes, this information should be described in the figure legend. Importantly, authors did not present TNF-gamma results in ELIPSOT assay, in which was described in materials and methods.

Q7: minor questions, spelling wrong “florescent” such as line 278, cDMEM (Line 147) should be DMEM.   

In summary, authors have a good idea to generate non-infectious virus, however,  the main issue of this manuscript is that too limited experiments to support their conclusions, especially the results that the Ad7 with fiber deletion don’t produce progeny virus in non-complement cells”. The conclusion is against previously published by Puvion-Dutileul et al (1999). Puvion-Dutileul et al demonstrated that the fiber protein is not essential for the assembly of progeny viruses. Therefore, I would suggest authors to provide more powerful experimental data to support their conclusions (Biology of the cell, 1999).

Reviewer 2 Report

The au have presented an essential paper based on the concept that the    fiberless adenovirus particles described by von Seggern et al 1999 and Legrand et al 1999 that were developed to retarget adenovirus vectors for gene therapy instead  could be developed as a mode for vaccination.

This group  has recently presented   influenza vaccine  and an Ebola vaccine  based Ad5 and Ad6  capsids with defect IIIa peptide  that can not form progeny virus.

Now  single cycle scAd7 is here evaluated as as means  to create  a vaccine  against  the respiratory patogen adenovirus 7. Au demonstrates that they can induce both humoral and cellular immunity against Ad 7 in hDSG mice. Au demonstrate that scAd7 is more immunogenic  than a replication defective adenovirus.

Comments and questions: Fig 1  How much of the fiber gene was deleted.

Give the coordinates. Fig 2:  Why is the yield of  all constructs lower in  the 293-Ad7-Fiber cells?

154  rephrase   in each virus.

194  each splenocyte was exposed to 5000vp  how many vp can they  take 

before they are affected.

282 Can the hDSG mice support infection  ie growth of Ad 7.

Can fiberless adenovirus bind lymphocytes or other cells  and then internalize  via vertexcapsomer bindning to integrins?

Do Au intend to evaluate im, intranasal or an enteric route of immunization

to obtain protective immunity?

Robert Chanock  was indeed  proud  of the enterocoated capsids

a means to achieve protected mucsal immunity towards Ad 4 and Ad7.

Now infectous virus can be shed for weeks from  the vaccinated recruit.

This is the rational for all the work presented in this paper.

Safety is essential in assessment  of  new vaccine candidates.

Why was the Ad 7 prototype strain Gomen used in these extensive studies.

This  strain has to my knowledge  not been used to vaccinate  anyone .

The vaccine evaluated in millions of recruits  is  the Ad 7a  subtype  with a different genome